# Performance of the ImmuView and BinaxNOW assays for the detection of urine and cerebrospinal fluid *Streptococcus pneumoniae* and *Legionella pneumophila* serogroup 1 antigen in patients with Legionnaires' disease or pneumococcal pneumonia and meningitis

**Paul H. Edelstein**[1,2]*, **Charlotte Sværke Jørgensen**[3], **Leslie A. Wolf**[4]

1 Department of Pathology and Laboratory Medicine, Perelman School of Medicine, University of Pennsylvania, Philadelphia, Pennsylvania, United States of America, 2 Department of Medicine, University of Cambridge, Cambridge, United Kingdom, 3 Department of Virus & Microbiological Special Diagnostics, Statens Serum Institut, Copenhagen, Denmark, 4 Department of Pathology and Laboratory Medicine, University of Louisville, Louisville, Kentucky, United States of America

* paul.edelstein@pennmedicine.upenn.edu

## Abstract

The performances of the ImmuView *Streptococcus pneumoniae* (Sp) and *Legionella pneumophila* (Lp) urinary antigen test were compared to that of the BinaxNOW Sp and Lp assays, using frozen urine from 166 patients with Legionnaires' disease (LD) and 59 patients with pneumococcal pneumonia. Thirty Sp-positive or contrived cerebrospinal fluids (CSF) were also tested. Test specimens were collected and tested at different sites, with each site testing unique specimens by technologists blinded to expected results. No significant differences in test concordances were detected for the ImmuView and BinaxNOW assays for the Sp or Lp targets for urine from patients with pneumococcal pneumonia or LD when performance from both sites were combined. At one of two test sites the ImmuView Lp assay was more sensitive than the BinaxNOW assay, with no correlation between test performance and Lp serogroup 1 monoclonal type. Urines from six of seven patients with LD caused by *Legionella* spp. bacteria other than Lp serogroup 1 were negative in both assays. Both tests had equivalent performance for Sp-positive CSF. The clinical sensitivities for pneumococcal pneumonia were 88.1 and 94.4% for the ImmuView and Binax assays, and 87.6 and 84.2% for the Lp assays, respectively. Test specificities for pneumococcal pneumonia were 96.2 and 97.0% for the ImmuView and Binax assays, and 99.6 and 99.1% for the Lp assays. Both assays were highly specific for Sp in pediatric urines from children with nasopharyngeal colonization by the bacterium. ImmuView and BinaxNOW assay performance was equivalent in these studies.

**Data Availability Statement:** All relevant data are within the manuscript and its Supporting Information files.

**Funding:** This study was supported by SSI Diagnostica (https://www.ssidiagnostica.com/) in the form of a salary provided to PHE. This salary support, which was required and administered by the University of Pennsylvania, amounted to 1% of his annual salary, and was used for study planning, study supervision, data analysis from all three study sites and writing the manuscript. The specific role of this author is articulated in the 'author contributions' section. SSI Diagnostica paid for the study reagents and supplies at each study site. Additionally, at the University of Pennsylvania and University of Louisville sites, SSI Diagnostica paid for the performing technologist's time. SSI Diagnostica reviewed the manuscript for accuracy of the reported data. The funders had no further role in data analysis, decision to publish, or preparation of the manuscript.

**Competing interests:** The authors have read the journal's policy and have the following competing interests: SSI Diagnostica (https://www.ssidiagnostica.com/) provided a salary to PHE. This salary support, which was required and administered by the University of Pennsylvania, amounted to 1% of his annual salary, and was used for study planning, study supervision, data analysis from all three study sites and writing the manuscript. He received no other support from SSI Diagnostica, prior to or after this study. Pernille E. Landsbo and Sanne Otte of SSI Diagnostica wrote the clinical trial protocols, provided logistic support and trained site technologists in the performance of the assays. Ida T. Andersen of SSI Diagnostica collated data from the study sites. Christopher Bentsen, an independent consultant hired by SSI Diagnostica, assisted in clinical trial protocol development and human subject committee applications. SSI Diagnostica reviewed the manuscript for accuracy of the reported data. This does not alter our adherence to PLOS ONE policies on sharing data and materials. There are no patents, products in development or marketed products associated with this research to declare.

## Introduction

Detection of antigenuria is a commonly used test for pneumococcal pneumonia and Legionnaires' disease, and in the case of the latter is the most common method of diagnosis of the disease[1, 2]. A number of commercial assays are used for these tests, with generally comparable performances, using both conventional enzyme immunoassays and lateral flow immunoassays. The tests for Legionnaires' disease are highly specific and are most sensitive for the detection of *L. pneumophila* serogroup 1, with only rare positive results for disease caused by *L. pneumophila* serogroups other than serogroup 1 or for other *Legionella* spp., except for a combination *L. pneumophila-L. longbeachae* lateral flow assay. In addition, some of the *L. pneumophila* serogroup 1 antigenuria tests are more sensitive for the Pontiac *L. pneumophila* serogroup 1 monoclonal group than for non-Pontiac strains[3]. Apart from the use of these assays to assist in the etiologic diagnosis of pneumonia, some of these assays can be used to detect pneumococcal antigen in cerebrospinal fluid to assist in the detection of pneumococcal meningitis[4].

The SSI Diagnostica ImmuView assay is a CE-marked and U.S. FDA-cleared lateral flow assay that detects both *L. pneumophila* and *S. pneumoniae* antigen using a single test strip (www.immuview.com). The Abbott BinaxNOW card assays for *L. pneumophila* and *S. pneumoniae* are U.S. FDA cleared devices that use separate test kits and test strips for each bacterium (www.alere.com/en/home/products-services/brands/binaxnow.html). We performed a comparison of these two test types to serve as data for a U.S. FDA 510k application for the ImmuView assay. These assays were carried out at one diagnostic laboratory in the USA and one in Denmark using a large number of archived frozen urine specimens specific to each laboratory, from patients with pneumococcal pneumonia or Legionnaires' disease, or neither disease. In addition, two U.S. laboratories and one Danish laboratory tested both test kits for the presence of pneumococcal antigen in CSF specimens. We show that both the ImmuView and BinaxNOW assays have equivalent performance.

## Materials and methods

Testing was performed at three study site laboratories, the Infectious Diseases Laboratory of the University of Louisville, the Clinical Microbiology Laboratory of the University of Pennsylvania Hospital (UPenn) and the serological laboratory of the Statens Serum Institut in Copenhagen (SSI). The SSI is a different entity, and independent from, SSI Diagnostica, the manufacturer of the ImmuView urine antigen test kit. Urines from patients with Legionnaires' disease and bacteremic pneumococcal pneumonia were tested at UPenn and SSI, with the specimens being different at the two sites. The 55 positive urine specimens from patients with Legionnaires' disease tested by SSI were novel and not previously used in a prior published evaluation of the ImmuView or any other test[5]. All patients with a diagnosis of Legionnaires' disease or bacteremic pneumococcal pneumonia had clinical findings consistent with acute pneumonia including pulmonary infiltrates on chest roentgenography. All patients with previously negative tests for Legionnaires' disease or pneumococcal pneumonia were suspected of having bacterial pneumonia by their ordering clinicians. The urine specimens tested were not the result of consecutive sampling during infection, were not selectively collected because of specific clinical findings, and were collected during the initial presentation of the pneumonia with only one specimen collected per patient.

Urines from 56 children with lower respiratory infection not due to pneumococcal pneumonia and known *S. pneumoniae* nasopharyngeal colonization status, were tested at SSI only. The criteria for pneumonia were clinical signs and symptoms of pneumonia. All children had respiratory specimens submitted for culture; none grew *S. pneumoniae*, while 12 grew

**Table 1. Specimens tested at each testing site.**

| Test site | Specimen type | | | |
|---|---|---|---|---|
| | CSF | Urine | | |
| | Pneumococcal meningitis[a] | Legionnaires' disease[b] | Pneumococcal pneumonia[c] | LRI children[d] |
| **UPenn** | 10/100[e]; 5/94[f] | 111/49[e]; 111/49[f] | 7/153[e]; 7/153[f] | not tested |
| **Univ of Louisville** | 10/100[e]; 0/0[f] | not tested | not tested | not tested |
| **SSI** | 10/100[e]; 8/88[f] | 55/183[e]; 54/178[f] | 52/186[e]; 47/184[f] | 0/56[g]; 0/55[h] |
| **All sites** | 30/300[e]; 13/182[f] | 166/232[e]; 165/227[f] | 59/339[e]; 54/337[f] | 0/56[g]; 0/55[h] |

[a] CSF from five patients with pneumococcal meningitis or five normal CSFs seeded with *S. pneumoniae*. All three study sites tested different CSF specimens, both positive and negative

[b] positive *Legionella* spp. sputum culture, prior positive *L. pneumophila* urine antigen assay, or both; negative controls were patients with prior negative *L. pneumophila* urine antigen tests, positive blood cultures for another pathogen (including *S. pneumoniae*), sputum cultures yielding another pathogen, or negative blood cultures and no respiratory pathogen isolated

[c] pneumococcal pneumonia with *S. pneumoniae* bacteremia; negative controls were from patients with other infections, including Legionnaires' disease

[d] children suspected of bacterial pneumonia, 13 of whom had positive nasopharyngeal cultures for *S. pneumoniae*. This group was included to study tests for *S. pneumoniae* and *L. pneumophila* antigenuria specificity only in children

[e] total positive/total negative patient urines or CSF specimens tested using ImmuView assay

[f] total positive/total negative patient urines or CSF specimens tested using BinaxNOW assays

[g] total positive/total negative patient urines tested using ImmuView assay and BinaxNOW *S. pneumoniae* assay

[h] total positive/total negative patient urines using BinaxNOW *L. pneumophila* assay

*Staphylococcus aureus*, three *Branhamella catarrhalis*, one *Acinetobacter* spp., and the rest normal upper respiratory flora. Seven of the children had positive prior testing on their urines for pneumococcal antigen using the Binax NOW *S. pneumoniae* antigen card, with the remainder being negative. All but four of the children had nasopharyngeal cultures obtained to determine *S. pneumoniae* colonization on the same day that urines were collected for antigen testing, with 13 children found to be colonized. Culture and prior pneumococcal antigen testing results were not used to select these specimens, with the exception that no child was included if the respiratory tract culture grew *S. pneumoniae*. The age of the children ranged from <1 to 19 years, with a median of 5.5 years and mean of 6.6 years. All three laboratories tested negative control cerebrospinal fluid (CSF), CSF from five patients with culture-confirmed pneumococcal meningitis and five contrived *S. pneumoniae*-containing CSF. One hundred CSF specimens were distributed for testing to each study site by SSI Diagnostica, with each site being sent different CSF specimens (Table 1). All specimens tested at each site were randomized and identified only by a number code, with the technologists performing testing being unaware of the true identity or expected result for each specimen. The urine specimens that were tested at each site had been stored frozen (-20 to -80 C) for up to 39 years after initial collection, and were all collected from adult patients with suspected pneumonia and known results for either *L. pneumophila* urine antigen tests, respiratory cultures for conventional bacterial pathogens, respiratory cultures for *Legionella* spp., blood cultures, or a combination of one or more of these tests (Table 2).

The BinaxNOW S. *pneumoniae* and *L. pneumophila* lateral flow assays, which are two separate tests, were performed according to the manufacturer's instructions. The ImmuView S. *pneumoniae* and *L. pneumophila* lateral flow assay, which is a single test, was also performed according to the manufacturer's instructions. Briefly, the BinaxNOW assay is performed by inserting a specimen-soaked kit-supplied swab onto a specimen pad on a card, followed by the addition of kit-supplied buffer to a 2nd well. The card is closed and incubated for up to 15 minutes at room temperature. Colored lines on the card indicate both internal control and target

**Table 2. Laboratory findings for patients with Legionnaires' disease.**

| Site | Culture positive Lp1[a] | | | Culture positive not Lp1 | Culture not done | | | Total |
|---|---|---|---|---|---|---|---|---|
| | UAg+[b] | UAg- | UAg Equiv+ | UAg neg | UAg + | UAg Equiv +, Outbreak Suspect | UAg Equiv + | |
| **UPenn** | 44 | 0 | 0 | 7[c] | 54 | 3 | 3 | 111 |
| **SSI** | 19 | 28 | 1 | 0 | 0 | 0 | 0 | 55 |

[a] Lp1, *L. pneumophila* serogroup 1

[b] UAg, urine antigen test positive (+), negative (-), or borderline positive (Equiv+)

[c] *L. wadsworthii*, *L. bozemanae*, *L. longbeachae* serogroup 1 (2 ea), *L. pneumophila* serogroup 2, *L. pneumophila* serogroup 4 (2 ea)

positivity. A positive result can be read at any time up to the 15 minute maximum incubation time and a negative result is read at the 15 minute mark. Any visible line is considered positive, regardless of the color strength or line width, as long as the control line is also positive.

The ImmuView assay is performed by adding a specified amount of urine (120μL or three drops) to a kit-supplied test tube containing a specified amount of buffer (90μL or two drops). The ImmuView strip is immersed in the test tube, removed after 15 minute's incubation at room temperature, and read while still wet. Three line positions can be read on the strip, the control line, a *L. pneumophila* line and a *S. pneumoniae* line, each of which are distinguished by a different color. Any line is considered positive regardless of color intensity as long as the line width is complete, with a requirement that the control line be positive for a valid assay. Partial width lines and colored spots are interpreted as invalid, requiring retesting of the sample in a new test.

Tests on the same specimen using different kits were not performed simultaneously or immediately sequentially to avoid reading bias. At least four hours separated the performance of each test type to help avoid reading bias, and if possible, a different technologist performed each test type on the specimen on a given day to further reduce reading bias. Results were recorded manually for each run, with a different recording sheet being used in the morning and afternoon readings for blinding purposes. Digital photographs of test strips and cards were made and stored after each run, to allow confirmation of the accuracy of the recorded results.

After both test types were performed, and the results recorded, the results were reviewed by a person not performing the testing to determine if the ImmuView and BinaxNOW results were discordant. If that were the case, then repeat testing of the specimen using the ImmuView and the target-specific BinaxNOW assay was performed, after boiling the urine for 10 minutes, centrifugation (~10,00 RCF) for one minute, and testing the supernate. The repeat testing was carried out one or more days after initial testing, with testing of the BinaxNOW and Immu-View assay separated by at least four hours, to reduce reading bias. Boiling and centrifugation was designed to reduce false-positive results due to heavy urine sediment and rheumatoid-like factors.

Calculations of test concordances were based solely on test results without regard to the known prior microbiology results. Because this trial was designed to determine the relative abilities of the two assays for *L. pneumophila* to detect Legionnaires' disease caused by confirmed *L. pneumophila* serogroup 1 infection, testing results of urines from patients with *Legionella* spp. infection other than *L. pneumophila* serogroup 1 or those whose initial urine antigen results were borderline positive were excluded from the primary analysis of test agreements and clinical sensitivity and specificity. A confirmed case of pneumococcal pneumonia required a positive blood culture for *S. pneumoniae*. Confirmed cases of Legionnaires' disease required either a positive respiratory tract culture for *Legionella* spp., a positive urine antigen assay for

*L. pneumophila*, or both positive tests. Prior microbiology laboratory results for each patient were unblinded only after the trial was completed.

Each laboratory was instructed in the use of the ImmuView assay by SSI Diagnostica. Prior to testing the unknown specimens, each technologist performing the testing successfully completed extensive proficiency and reproducibility examinations using blinded specimens, and was also observed for proper technique by a SSI Diagnostica expert in use of the test. Instruction in the proper performance of the BinaxNOW tests was given by a different expert in the use of these tests, with technologists not allowed to perform testing on unknown testing until proper test performance was ascertained by observation and by completion of a panel of positive and negative specimens with complete accuracy. Thus, the participating technologists were highly trained, and accurately performed testing, prior to testing of the unknown clinical specimens.

Statistical testing of differences for the paired sample agreement assays were performed using a two-tailed McNemar test[6]. Analysis of clinical sensitivity and specificity uses standard definitions[7], for which 95% confidence intervals were calculated using the modified Wald method [8]. Comparisons of clinical sensitivity and specificity between sites used the two-tailed Fisher exact test (GraphPad InStat3.10).

The human studies protocols of all three study sites were approved by the Western Institutional Review Board (IRB) for human studies, protocol 20162534. In addition, the Western IRB reviewed and approved an additional protocol for the SSI site, protocol 1170324. The University of Pennsylvania Human Studies IRB accepted the Western IRB approval of the test protocol under protocol number 10060242. The University of Louisville Human Studies IRB approved the study under protocol number 16.0531. Patient consents were not obtained because these were not required by the IRBs.

The study sponsor, SSI Diagnostica, wrote the test protocols, paid for the study reagents and supplies, as well as for the performing technologist's time at all sites except for the SSI site. This report was written independently of SSI Diagnostica using data spreadsheets supplied by the sponsor as well as separately kept spreadsheets for the UPenn site. SSI Diagnostica reviewed the manuscript for accuracy of the reported data but otherwise had no role in its writing. All authors vouch for the accuracy of this report.

## Results

Urines from 166 and 59 patients with Legionnaires' disease and bacteremia pneumococcal pneumonia, respectively, were tested. Thirty actual or contrived *S. pneumoniae*-positive cerebrospinal fluids (CSF) and 56 urines from children were tested (Table 1).

When testing for bacteremic pneumococcal pneumonia the ImmuView and BinaxNOW assays had near perfect agreement (Table 3). Clinical sensitivities for these assays were 88.1 and 94.4% for the ImmuView and BinaxNOW assays, respectively, with overlapping 95% confidence intervals (Table 4). Test specificities were 99.6 and 97.0% for the ImmuView and BinaxNOW assay, respectively. Testing site-specific test sensitivity, but not specificity, was significantly lower for the UPenn than the SSI site for both assay types (P<0.01 by Fisher's exact

**Table 3. *S. pneumoniae* specimens test agreements for urines from adults.**

|  | BinaxNOW | |
|---|---|---|
| **ImmuView** | **positive** | **negative** |
| **positive** | 53 | 7 |
| **negative** | 8 | 323 |

p = 1.0, McNemar test

**Table 4. Clinical sensitivity and specificity for urines from adults.**

| Assay | Target | Sensitivity[a] (%) | Specificity[b] (%) |
|---|---|---|---|
| **ImmuView** | *S. pneumoniae* | 88.1 (77.1 to 94.3), 52/59 | 96.2 (93.5 to 97.8), 326/339 |
| **BinaxNOW** | | 94.4 (84.2 to 98.6), 51/54 | 97.0 (94.5 to 98.4), 327/337 |
| **ImmuView** | *L pneumophila* | 87.6 (81.3 to 91.9), 134/153 | 99.6 (97.3 to 100), 231/232 |
| **BinaxNOW** | | 84.2 (77.5 to 89.2), 128/152 | 99.1 (96.6 to 99.9), 225/227 |

a, mean (95% CI), total test positive patients with infection/total patients with infection specific to target

b, mean (95% CI), total test negative patients without infection/total patients without infection specific to target

test) (S1 Table). These site specific differences in test sensitivity were not reflected in test agreements for the two different assays for each study site (S2 and S3 Tables), where no statistically significant differences were found by McNemar tests.

The ImmuView *L. pneumophila* test detected nine positive urines that were BinaxNOW-negative versus five BinaxNOW-positives that were ImmuView-negative (p = 0.42) (Table 5). This imbalance in test sensitivity was solely due to a non-statistically significant difference (p = 0.11) in test concordance at the SSI site, but not the UPenn site (S4 and S5 Tables). These site-specific test agreement differences between the BinaxNOW and ImmuView assays had the same trend, albeit with overlapping confidence intervals when the data were analyzed for clinical sensitivity. No apparent difference was detected between sites for clinical specificity (S6 Table). The ImmuView and BinaxNOW assays had 92 and 94% sensitivity, respectively, at the UPenn site, but 80 and 67% sensitivity for the SSI site, for the ImmuView and BinaxNOW assays, respectively (UPenn vs. SSI p = 0.001 for comparison of sensitivity for each test type, Fisher's exact test). Because test sensitivity has been reported to be higher for *L. pneumophila* serogroup 1 Pontiac monoclonal types, we analyzed the SSI site results (the only site having monoclonal typing results, for the 48 typed isolates) for BinaxNOW-ImmuView agreements by monoclonal antibody type (S7 Table). There was no association between BinaxNOW-ImmuView discordance and monoclonal antibody type (p = 0.2, Fisher's exact test). We analyzed the effect of monoclonal antibody type on the seven specimens with available strain typing data that were falsely-negative for *L. pneumophila* by BinaxNOW assay, and truly-positive by ImmuView assay at the SSI site (S8 Table). Five of the seven urines were obtained from patients with Legionnaires' disease caused by the Pontiac monoclonal group, demonstrating no correlation between Pontiac monoclonal group and test sensitivity differences between the two assays. In addition, for all typed infection specimens regardless of discrepancies, ImmuView sensitivity for Pontiac and non-Pontiac infections was 87 and 68%, respectively, and BinaxNOW sensitivity was 65 and 60% for Pontiac and non-Pontiac infections, respectively (p>0.1 for comparisons within or between tests, Fisher's exact tests).

Not included in the calculations of test performance for the *L. pneumophila* assays were urines from 13 UPenn patients, either because the patients had Legionnaires' disease not caused by *L. pneumophila* serogroup 1, or because they had only borderline-positive

**Table 5. *L. pneumophila* test agreements for urines from adults.**

| | BinaxNOW | |
|---|---|---|
| **ImmuView** | **positive** | **negative** |
| **positive** | 125 | 9 |
| **negative** | 5 | 240 |

p = 0.42, McNemar test

("equivocal") urine antigen test results, using either a commercial radioimmunoassay or ELISA, without positive cultures at the time of initial testing. Seven patients had culture-positive Legionnaires' disease caused by *L. wadsworthii*, *L. bozemanae*, *L. longbeachae serogroup* 1 (2 patients), *L. pneumophila* serogroup 2 and *L. pneumophila* serogroup 4 (2 patients). Six of the seven urines from these patients were not detected as positive by either assay, and one from a patient with *L. pneumophila* serogroup 4 disease was detected by both assay types. Three patients were suspected of epidemic Legionnaires' disease and were investigated as outbreak subjects during two different Legionnaires' disease outbreaks, both caused by *L. pneumophila* serogroup 1, Pontiac monoclonal group; none of these three patients had positive cultures and all three had borderline positive urine antigen assays when they were first tested decades previously. When retested during this trial two of the outbreak-related urines were positive, but only in the ImmuView assay, with the third being negative in both assays. Three additional urines were tested that were taken from patients with suspected sporadic Legionnaires' disease, with the only laboratory test indication of the disease being borderline-positive urine antigen tests at the time of original collection; one of these three was positive in both the BinaxNOW and ImmuView assays, with the other two being negative in both assay types. Inclusion of the data from these 13 urines into tests of agreement resulted in no significant difference for the UPenn site (p = 1.0, McNemar test) (S9 Table), and a change in the UPenn test sensitivity, but not specificity, with the assumption that all 13 of these patients truly had Legionnaires' disease. Test sensitivities for the UPenn site changed to 85.0% for both the ImmuView and BinaxNOW assays, respectively (S6 Table footnote). When these data are included in overall test performance for both the SSI and UPenn sites test agreements were not significantly different (p = 0.21 by McNemar test) with a change in test sensitivities to 83.1 and 79.4% for the ImmuView and BinaxNOW assays, respectively (S10 and S11 Tables).

Testing of CSF specimens from patients with pneumococcal meningitis, or contrived-positive CSF specimens, showed almost perfect agreement between BinaxNOW and ImmuView *S. pneumoniae* tests (p = 1, McNemar test) (S12 Table). Because the positive CSF specimens were comprised both of CSF specimens from patients with pneumococcal meningitis, and contrived positive CSF specimens, clinical sensitivity and specificity were not calculated, but rather positive and negative agreement. Positive agreement was high for both assays in comparison with the known result, with overlapping 95% confidence intervals, with values of 96.7 and 92.3% for the ImmuView and BinaxNOW assays, respectively (S13 Table). Both the BinaxNOW and ImmuView *S. pneumoniae* assays exhibited high negative agreement in comparison with the known result (98.9 and 97.7%, respectively) (S13 Table). The ImmunView *L. pneumophila* component of the dual target test strip, which is not indicated for testing of CSF, exhibited high negative agreement of 98.8% (95% CI 96.8 to 99.7%). No CSF testing was performed using the BinaxNOW *L. pneumophila* assay.

Testing of urines from children with lower respiratory tract infection, none of whom had pneumococcal pneumonia based on the absence of a positive culture result for *S. pneumoniae*, was performed to determine the specificity of urine antigen testing since nasopharyngeal pneumococcal colonization is known to lead to falsely-positive urine antigen assays. Thirteen of the 56 children tested for had pneumococcal colonization of the nasopharynx. Three of 56 urines were positive by both the BinaxNOW and ImmuView *S. pneumoniae* assays, and two urines were BinaxNOW-positive but ImmuView-negative (p = 0.5, McNemar test) (S14 Table). One of the three ImmuView-positive urines was from a child with pneumococcal colonization, as were two of the five BinaxNOW-positive tests. No BinaxNOW or ImmuView *L. pneumophila* test was positive for these urine samples (S15 Table). Clinical specificity for the *S. pneumonia*e target was 94.6 and 91.1% for the ImmuView and BinaxNOW assays, respectively, with overlapping 95% confidence intervals (S16 Table).

Since false-positive urine antigen tests have been reported as the result of rheumatoid-like factors in the urine that are inactivated by boiling, urine specimens that gave discrepant results between BinaxNOW and ImmuView assays were retested after boiling and centrifugation[9]. This was done only if there was sufficient specimen available to conduct a test, with priority given to the ImmView assay if there was insufficient urine to conduct both ImmuView and BinaxNOW assays. Priority was given to testing the ImmuView in the case of insufficient volume for both tests because less was known about the effect of specimen boiling on ImmuView results than is known about this effect on Binax assays. Sufficient volume was available at the UPenn site for retesting of all discrepant pairs, but for the SSI site only one of the 18 discrepant pairs were tested using both the ImmunView and BinaxNOW assays (S17 Table). The results after this treatment were not used to determine test agreements or clinical sensitivity and specificity. For the three BinaxNOW-ImmuView discordant UPenn *L pneumophila* results for urines from patients with Legionnaires' disease, only one discrepancy was resolved after boiling treatment. For all three UPenn urines from patients with Legionnaires' disease that tested positive with either the BinaxNOW or ImmuView *S. pneumoniae* assay, boiling resolved the discrepancies. One of two UPenn urines from patients with bacteremic pneumococcal pneumonia changed to a concordant BinaxNOW-ImmuView *S. pneumoniae* result.

Some of the results of boiling treatment were unexpected because three ImmuView *S. pneumoniae* truly-negative specimens became falsely-positive after treatment and some falsely-negative specimens became true-positives for both test types, as well as the expected false-positives becoming true-negative (S17 Table). For example, three of ten ImmuView-true positive but BinaxNOW-false negative *L. pneumophila* specimens reverted to negative after boiling; unfortunately, there was inadequate urine available to perform a test with the BinaxNOW assay on these boiled specimens. The three ImmunView true-positive to false-positive specimens were weakly positive prior to boiling based on the intensity of the positive band. Because many discordant pairs had post-boiling testing performed only using the ImmuView assay, comparisons of the frequencies of the various outcomes between BinaxNOW and ImmuView assays cannot be determined.

## Discussion

This study showed that the ImmuView and BinaxNOW assays had excellent test agreements and clinical performances for both the *S. pneumoniae* and *L. pneumophila* targets for both urine and CSF specimens. The very large number of urines that were tested from patients with confirmed Legionnaires' disease provided a robust estimate of assay performance, as did the use of several hundred negative controls to determine assay specificities. Use of specimens from more than one site showed that relative assay performance differed between sites for the *L. pneumophila* target, suggesting the possibility of differing performance based on strain type or clinical severity.

There was very high test concordance between the two assays for the *S. pneumoniae* target for urine specimens from patients with pneumococcal bacteremia, with only two discordant results. Comparing these results with the two previously published studies of the ImmuView vs. BinaxNOW assays for the detection of *S. pneumoniae* antigenuria is impossible, because none of the prior studies revealed concordance data, only sensitivity and specificity rates[10, 11]. Jørgensen and colleagues reported that the ImmuView and BinaxNOW *S. pneumoniae* assays were 85 and 77% sensitive, respectively, for patients with pneumococcal bacteremia, and 98.7 and 100% specific[10]. Athlin and colleagues reported sensitivities of 60 and 62% for the BinaxNOW and Immunview tests, respectively, with 97% specificity for both tests[11]. These data contrast with the 88 and 94% sensitivities found by us, for the ImmuView and

BinaxNOW assays, respectively, with respective specificities of 93 and 97%. The lower sensitivity of the ImmuView assay than the BinaxNOW assay at the UPenn site represents a difference of two results out of only seven total known positive urines tested, and is unlikely to represent a truly significant difference in assay performance. Based on our results, plus those of the two published studies, there are insignificant differences between the performances of these two tests for the detection of *S. pneumoniae* antigenuria.

Evaluation of the performances of the two different assays for the detection of *L. pneumophila* antigenuria was more complicated than for the *S. pneumoniae* target because of testing site differences in relative clinical sensitivity. Test sensitivities, but not specificities, were significantly greater at the UPenn site than at the SSI site. We were unable to explain this based on strain monoclonal antibody typing data from the SSI site. Another possibility for this difference is that 54% of the UPenn positive specimens tested were from antigenuria-positive only specimens, perhaps resulting in a selection bias. Whether other factors such as patient or unmeasured bacterial strain differences also accounted for these differences is unknown. Both test types had non-significantly different test result concordances within each site, and for the combined sites, good evidence for equivalent performance of these two test types for the detection of *L. pneumophila* serogroup 1 antigenuria.

In a previously published different study performed at the SSI, using different specimens, the ImmuView *L. pneumophila* assay detected 88 of 99 (89%) *L. pneumophila* serogroup 1 culture-positive or urine antigen-positive specimens, versus 71 of 99 (72%) specimens for the BinaxNOW assay[10]. For urines from solely *L. pneumophila* serogroup 1 culture-positive patients, these numbers were 48 of 55 (87%) and 43 of 55 (78%) positive tests for the ImmuView and BinaxNOW assays, respectively; without test concordance data it is impossible to perform appropriate statistical testing on these paired specimens, but the ImmuView assay appeared to be more sensitive in that study.

In the previously published SSI study both the BinaxNOW and ImmuView tests were better able to detect Legionnaires' disease caused by the *L. pneumophila* serogroup Pontiac monoclonal group than the non-Pontiac monoclonal groups, although these differences were not significantly different [10]. We observed the same differences in test sensitivity depending on monoclonal group, which were also not statistically significantly different between or within test types.

Neither the ImmuView nor the BinaxNOW *L. pneumophila* assays were able to detect urine antigen from six of the seven patients with Legionnaires' disease caused by non-*L. pneumophila* serogroup 1 bacteria, with both detecting one of the two *L. pneumophila* serogroup 4 infections. The inability to detect urine antigen from the four patients with Legionnaires' disease caused by *Legionella* spp. other than *L. pneumophila* was expected, has been reported previously[10], and is not claimed in the product insert of either test. A prior study showed that the ImmuView test detected 0/2, 0/5, 2/14, 10/28 and 1/1 of urines from patients with *L. pneumophila* serogroup 4, 5, 6, 3 and 15 infections, with the BinaxNOW assay only detecting 1 of 28 serogroup 3 urines[10]. We add to this the inability of either assay to detect urine antigen from a patient with *L. pneumophila* serogroup 2 infection.

Our test of CSF specimens showed excellent concordance between test types. Although the small number of positive specimens tested results in a broad estimate of test sensitivity, test performance for both tests parallels that found in large clinical studies for the BinaxNOW assay[4, 12], lending credence to our findings.

The use of urine antigen detection for the diagnosis of pediatric pneumococcal pneumonia is non-specific because 25 to 50% of children with pneumococcal nasopharyngeal colonization have been reported to have positive antigen test results[13, 14]. We found a much lower fraction of positive tests for either test type in colonized children, 5 to 9%. The reason for this

difference is unknown, but regardless the specificities of each test were not significantly different for pediatric urine specimens.

Confirming a positive urine antigen assay by retesting the urine after boiling and centrifugation is commonly used to exclude false-positive tests due to rheumatoid-like factors, and for some tests such as the ImmuView test, this procedure is found in the kit instructions[15, 16]. Several studies have reported no change in test sensitivity using this procedure[9–11, 17]. It was therefore surprising that boiling and centrifugation of discrepant specimens led not only to falsely-positive tests becoming truly-negative as expected, but also truly-positive urines becoming falsely-negative. In addition, some urine specimens converted from falsely-negative to truly-positive. Since many of these urines were retested only with the ImmuView test, the greater frequency of these unexpected reactions with the ImmuView test does not necessarily point to this being more common with that test. It is possible that these aberrant results were due to prior freezing of the urines, to antigen levels very close to the detection limit or to some other factor. It would be wise to determine the frequency of such results in a prospective study for the both tests with both negative and positive freshly collected urines.

Several potential limitations of this study should be mentioned. First, positive urines for testing were unavoidably selected on the basis of a prior positive urine antigen test or a positive respiratory (*Legionella* spp.) or blood culture (*S. pneumoniae*). This results in a selection bias[18], because it was highly likely that these preselected urines would be found positive in the test assays. Thus our estimates of test sensitivity are likely an overestimate of the true sensitivity in patients without known prior positive tests or a high severity illness. Test specificity should not be affected in this case. There is also a likely spectrum bias, because urines from patients with pneumococcal pneumonia and bacteremia, and patients with positive respiratory cultures for Legionnaires' disease, were overrepresented in our test population. Therefore, comparative test agreement, rather than clinical sensitivity, is the best measure of the relative performances of these two different test kits. Another potential limitation is that all of the tested urines had been previously frozen, some several decades previously. While the antigen targets are relatively stable, there can be degradation of the antigen with prolonged storage[19]. This would have the opposite effect of the selection bias, in that urines from patients with Legionnaires' disease or pneumococcal pneumonia may have lost sufficient antigen to become negative and produce a falsely-negative test result.

In conclusion, both the ImmuView *S. pneumoniae* and *L. pneumophila* Urinary Antigen Test, and the BinaxNOW *S. pneumoniae* and *L. pneumophila* antigen cards for the same organisms performed equally well for the detection of Legionnaires' disease caused by *L. pneumophila* serogroup 1, bacteremic pneumococcal pneumonia and pneumococcal meningitis. Regional variations in *L. pneumophila* strains causing Legionnaires' disease may affect the relative performance of these tests but further delineation of this in prospective studies is required.

## Supporting information

**S1 Table. Clinical sensitivity and specificity for *S. pneumoniae* antigenuria by study site.**
(PDF)

**S2 Table. Agreement of *S. pneumoniae* antigenuria testing, UPenn.**
(PDF)

**S3 Table. Agreement of *S. pneumoniae* antigenuria Testing, SSI.**
(PDF)

**S4 Table. Agreement of L. pneumophila antigenuria testing Lp SG1 only, UPenn.**
(PDF)

**S5 Table. Agreement of *L. pneumophila* antigenuria testing, SSI.**
(PDF)

**S6 Table. Clinical sensitivity and specificity for *L. pneumophila* antigenuria by study site.**
(PDF)

**S7 Table. Correlation of BinaxNOW and ImmuView *L. pneumophila* results and monoclonal group for concordant and discordant results, SSI.**
(PDF)

**S8 Table. Correlation of BinaxNOW *L. pneumophila* falsely-negative tests with monoclonal group, SSI site.**
(PDF)

**S9 Table. Agreement for UPenn site only of BinaxNOW and ImmuView *L. pneumophila* assays including urines from patients with *Legionella* infections other than non-*L. pneumophila* serogroup 1, culture-negative outbreak suspects and culture-negative sporadic cases with initial borderline-positive urine antigen tests.**
(PDF)

**S10 Table. Agreement of BinaxNOW and ImmuView L. pneumophila assays including urines from UPenn patients with *Legionella* infections other than non-*L. pneumophila* serogroup 1, culture-negative outbreak suspects and culture-negative sporadic cases with initial borderline-positive urine antigen tests[a]. SSI and UPenn sites combined.**
(PDF)

**S11 Table. BinaxNOW and ImmuView *L. pneumophila* clinical sensitivity and specificity when including specimens from patients with *Legionella* infections other than non-*L. pneumophila* serogroup 1, culture-negative outbreak suspects and culture-negative sporadic cases with initial borderline-positive urine antigen tests. SSI and UPenn combined data.**
(PDF)

**S12 Table. *S. pneumoniae* CSF agreements.**
(PDF)

**S13 Table. Positive and negative agreement for CSF specimens.**
(PDF)

**S14 Table. *S. pneumoniae* pediatric urines agreements.**
(PDF)

**S15 Table. *L. pneumophila* pediatric urines agreements.**
(PDF)

**S16 Table. Clinical specificity for pediatric urines.**
(PDF)

**S17 Table. Effects of retesting urine after boiling for ImmuView-BinaxNOW discordant result specimens for Upenn and SSI sites.**
(PDF)

## Acknowledgments

Pernille E. Landsbo and Sanne Otte of SSI Diagnostica wrote the clinical trial protocols, provided logistic support and trained site technologists in the performance of the assays. Ida T.

Andersen of SSI Diagnostica collated data from the study sites. Christopher Bentsen, an independent consultant hired by SSI Diagnostica, assisted in clinical trial protocol development and human subject committee applications. Martha Edelstein and Karina Rios performed testing at the University of Pennsylvania. Subathra Marimuthu, Laura G. Schindler, S. Sabrena Garr and Jennifer L. Wick performed testing at the University of Louisville, and James Summersgill initially directed testing there. Anders Frische, Brian Hauschildt, Suheil Nasim, and Regina Hansen performed testing at the Serological Unit, Statens Serum Institut.

## Author Contributions

**Data curation:** Paul H. Edelstein, Charlotte Sværke Jørgensen, Leslie A. Wolf.

**Formal analysis:** Paul H. Edelstein.

**Investigation:** Paul H. Edelstein, Charlotte Sværke Jørgensen, Leslie A. Wolf.

**Methodology:** Paul H. Edelstein, Charlotte Sværke Jørgensen, Leslie A. Wolf.

**Project administration:** Paul H. Edelstein, Charlotte Sværke Jørgensen, Leslie A. Wolf.

**Resources:** Paul H. Edelstein, Charlotte Sværke Jørgensen, Leslie A. Wolf.

**Supervision:** Paul H. Edelstein, Charlotte Sværke Jørgensen, Leslie A. Wolf.

**Validation:** Paul H. Edelstein, Charlotte Sværke Jørgensen, Leslie A. Wolf.

**Writing – original draft:** Paul H. Edelstein.

**Writing – review & editing:** Paul H. Edelstein, Charlotte Sværke Jørgensen, Leslie A. Wolf.

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
