## [Decision Letter · Decision Letter 0]

24 Jun 2020

PONE-D-20-15595

Performance of the ImmuView and BinaxNOW assays for the detection of urinary and cerebrospinal fluid Streptococcus pneumoniae and Legionella pneumophila serogroup 1 antigen in Legionnaires’ disease or pneumococcal pneumonia and meningitis

PLOS ONE

Dear Dr. Edelstein,

Thank you for submitting your manuscript to PLOS ONE. After careful consideration, we feel that it has merit but does not fully meet PLOS ONE’s publication criteria as it currently stands. Therefore, we invite you to submit a revised version of the manuscript that addresses the points raised below during the review process.

We look forward to receiving your revised manuscript.

Kind regards,

Ray Borrow, Ph.D., FRCPath

Academic Editor

PLOS ONE

Journal Requirements:

2. Please provide additional details regarding participant consent.

In the ethics statement in the Methods and online submission information, please ensure that you have specified (a) whether consent was informed;

(b) what type you obtained (for instance, written or verbal, and if verbal, how it was documented and witnessed). If the need for consent was waived by the ethics committee, please include this information;

(c) whether you obtained consent from parents or guardians of the children included in your study.

'PHE received funding from SSI Diagnostica to conduct this study

CSJ received funding from SSI Diagnostica to conduct this study

LAW received funding from SSI Diagnostica to conduct this study

SSI Diagnostica: https://www.ssidiagnostica.com/

The study sponsor, SSI Diagnostica, wrote the test protocols, paid for the study reagents and supplies as well as for the performing technologist’s time. This report was written independently of SSI Diagnostica using data spreadsheets supplied by the sponsor as well as separately kept spreadsheets for the UPenn site. SSI Diagnostica reviewed the manuscript for accuracy of the reported data but otherwise had no role in its writing.'

We note that you received funding from a commercial source: SSI Diagnostica

5. Please amend either the title on the online submission form (via Edit Submission) or the title in the manuscript so that they are identical.

Reviewers' comments:

Reviewer's Responses to Questions

**Comments to the Author**

1. Is the manuscript technically sound, and do the data support the conclusions?

Reviewer #1: Yes

Reviewer #2: Partly

2. Has the statistical analysis been performed appropriately and rigorously? 

Reviewer #1: I Don't Know

Reviewer #2: Yes

3. Have the authors made all data underlying the findings in their manuscript fully available?

Reviewer #1: Yes

Reviewer #2: Yes

4. Is the manuscript presented in an intelligible fashion and written in standard English?

Reviewer #1: Yes

Reviewer #2: Yes

5. Review Comments to the Author

Reviewer #1: Overall a good paper describing the results of a multisite study comparing the ImmunView and BinaxNOW urine antigen and CSF tests for Streptococcus pneumoniae and Legionella pneumophila. I would like to have seen more samples from patients with bacteraemic pneumococcal pneumonia tested at the UPenn site as I feel that the lack of these samples makes site comparisons for this particular test group difficult. I also have some minor points for you to consider.

1. Line 47- I don’t think it is necessary to describe the disease as “pneumonia caused by either pneumococcal pneumonia”. I would suggest something like “pneumococci” instead of “pneumococcal pneumonia” or alternatively remove the "caused by " and just list pneumococcal pneumonia.

2. Line 59- Please could you list the criteria that was used to define the children with lower respiratory infection not due to pneumococcal pneumonia. As cases of co-infections with pneumococci and other bacteria or viruses have been reported, was this considered when selecting these samples?

3. Line 172-174- The footnote for Table 1, specifically references e and f, are used to describe the total "urines" tested. However, the data in the first column of the table are supposed to represent CSF samples. Please can you clarify.

4. Line 265 - Please can you clarify as to what is meant by the ImmunView L. pneumophila test. I am assuming that it means that some of the samples reported a result for L. pneumophila when tested with the ImmunView test.

5. Line 283-286 -Please can you explain why priority was given to the ImmunView test over the BinaxNow when conducting repeat testing with insufficient sample volume.

6. Line 381-383 - Have you considered the sensitivity of the assays in regard to the different pneumococcal serotypes?

7. Table 4 - Could this table also include the figures for the number of samples reported positive and negative out of the total tested that was used to calculate the sensitivity and specificity?

Reviewer #2: In the present study, Edelstein et al compared the performances of ImmuView S. pneumoniae and L. pneumophila urinary antigen tests with that of the BinaxNOW S. pneumoniae and L. pneumophila assays using frozen samples and performing the testing at three laboratories. The study sponsor SSI Diagnostica wrote the test protocols and financed the study, but the report was written independently of them. The report adds information regarding the ImmuView test performance in comparison with the BinaxNOW assays on clinical samples.

My major concern is that samples are referred to as collected from pneumococcal pneumonia and Legionnaires’ disease patients but data are lacking on collection procedure and patient characteristics. When and where did the authors collect samples, and when were they analyzed? If the 55 Legionella serogroup 1 samples tested at SSI are the same 55 samples analyzed by Jorgensen et al (2015) previously this should be presented. Patient characteristics from those with suspected Legionnaires’ disease would help to interpret discordant results, especially since more than half of Legionnaires’ disease cases from the UPenn site are defined only by previous positive antigen tests (see below). It would be suitable to refrain from using the term “clinical” for defining test performance to avoid misunderstandings concerning the trial design.

Another concern is the lack of criteria for pneumonia diagnosis. Usually, a majority of patients with pneumococcal bacteremia have airway infections but, sometimes, may have meningitis, otitis or other foci only. If not described, maybe the authors should refer to patients with bacteremia and not bacteremic pneumonia. Furthermore, are these urine samples collected consecutively during certain time periods, or are they selected for some reason due to special features, e.g. if a positive pneumococcal antigen test was positive in the clinical routine along with positive blood culture? Could the selection of samples explain the high sensitivity yields for pneumococcal antigen testing for both tests in the study, and for site specific differences?

Line 115: How was serogroup 1 identified?

Line 118: How was pneumonia defined?

Line 155: Bacteremic pneumococcal pneumonia. Since the definition of Legionnaires’ disease and pneumococcal pneumonia is not clear, and based on positive microbiology results only, it should be referred to possible diagnoses.

Line 157: The sentence refers to Table 1 only.

Line 178: Supportive laboratory findings for patients with suspected Legionnaires disease?

Line 199: Not clinical performance, since the definitions of diagnoses have not yet been clarified.

Line 210: Not clinical specificity, since the definitions of diagnoses have not yet been clarified.

Line 261: Not clinical sensitivity, half the samples are spiked.

Line 269: Patient characteristics? Age? Previous urinary antigen test performed? Was urine sample and nasopharynx sample collected simultaneously, or colonization “at any time”?

Line 340: Is it 54% (60/111)? Please refer to Table 2 If appropriate.

Line 352: Due to absent test concordance…? Words missing.

Line 404: Little or no information of samples tested for specificity is presented in the study. Thus, if the samples are from healthy individuals, the results only reflects specificity on such a healthy population without respiratory symptoms. It would be of interest to study the test performances on patients with non-Legionella non-pneumococcal lower airway infections or bacteremia.

Line 408: Here, it is stated that the use of the term clinical sensitivity is not appropriate in the study, I agree.

Lines 388-391: If the selection of urine samples are based on positive Legionella urinary antigen tests only, with no clinical data on pneumonia, samples may be true-negative from the beginning, but false-positive before boiling. This is a major issue of the study, see above. Besides the discrepant specimens, could there be more positive tests that would have turned negative after boiling?

6. PLOS authors have the option to publish the peer review history of their article (what does this mean?). If published, this will include your full peer review and any attached files.

Reviewer #1: No

Reviewer #2: No

---

## [Author Response · Author response to Decision Letter 0]

31 Jul 2020

please see the rebuttal letter for these responses

---

## [Decision Letter · Decision Letter 1]

13 Aug 2020

PONE-D-20-15595R1

Performance of the ImmuView and BinaxNOW assays for the detection of urine and cerebrospinal fluid Streptococcus pneumoniae and Legionella pneumophila serogroup 1 antigen in Legionnaires’ disease or pneumococcal pneumonia and meningitis

PLOS ONE

Dear Dr. Edelstein,

Thank you for submitting your manuscript to PLOS ONE. After careful consideration, we feel that it has merit but does not fully meet PLOS ONE’s publication criteria as it currently stands. Therefore, we invite you to submit a revised version of the manuscript that addresses the final minor points raised during the review process.

We look forward to receiving your revised manuscript.

Kind regards,

Ray Borrow, Ph.D., FRCPath

Academic Editor

PLOS ONE

Reviewers' comments:

Reviewer's Responses to Questions

**Comments to the Author**

1. If the authors have adequately addressed your comments raised in a previous round of review and you feel that this manuscript is now acceptable for publication, you may indicate that here to bypass the “Comments to the Author” section, enter your conflict of interest statement in the “Confidential to Editor” section, and submit your "Accept" recommendation.

Reviewer #1: All comments have been addressed

2. Is the manuscript technically sound, and do the data support the conclusions?

Reviewer #1: Yes

3. Has the statistical analysis been performed appropriately and rigorously? 

Reviewer #1: I Don't Know

4. Have the authors made all data underlying the findings in their manuscript fully available?

Reviewer #1: Yes

5. Is the manuscript presented in an intelligible fashion and written in standard English?

Reviewer #1: Yes

6. Review Comments to the Author

Reviewer #1: Thank you for submitting the revised manuscript. I am satisfied with the authors response to the reviewers’ comments and believe that the manuscript should be accepted. I would however like to suggest a couple of minor changes for the authors to consider.

1. In lines 304-305 you state “Testing of urines from children with lower respiratory tract infection, none of whom had pneumococcal pneumonia, was performed to determine the specificity of urine antigen testing...”

A number of publications have highlighted that the diagnosis of pneumonia caused by pneumococci can be insensitive when based solely on culture. In fact, this is a reason in favour of the use of urine antigen diagnostic kits such as the BinaxNOW and ImmuView. For this reason, I believe that it is incorrect to state that these children did not have pneumococcal pneumonia and instead suggest that you re-word this sentence to note that the children were deemed not to have pneumococcal pneumonia based on the absence of any culture.

2. Lines 327-328 – “For the three BinaxNOW-ImmuView discordant result UPenn L pneumophila results for urines from patients with Legionnaires' disease…” – perhaps consider removing the first "result" from this sentence for clarification.

7. PLOS authors have the option to publish the peer review history of their article (what does this mean?). If published, this will include your full peer review and any attached files.

Reviewer #1: No

---

## [Author Response · Author response to Decision Letter 1]

13 Aug 2020

please see the cover letter and response to reviewer file

---

## [Editor Report · Decision Letter 2]

18 Aug 2020

Performance of the ImmuView and BinaxNOW assays for the detection of urine and cerebrospinal fluid Streptococcus pneumoniae and Legionella pneumophila serogroup 1 antigen in Legionnaires’ disease or pneumococcal pneumonia and meningitis

PONE-D-20-15595R2

Dear Dr. Edelstein,

We’re pleased to inform you that your manuscript has been judged scientifically suitable for publication and will be formally accepted for publication once it meets all outstanding technical requirements.

Kind regards,

Ray Borrow, Ph.D., FRCPath

Academic Editor

PLOS ONE
---

## [Editor Report · Acceptance letter]

20 Aug 2020

PONE-D-20-15595R2 

Performance of the ImmuView and BinaxNOW assays for the detection of urine and cerebrospinal fluid Streptococcus pneumoniae and Legionella pneumophila serogroup 1 antigen in patients with Legionnaires’ disease or pneumococcal pneumonia and meningitis 

Dear Dr. Edelstein:

I'm pleased to inform you that your manuscript has been deemed suitable for publication in PLOS ONE. Congratulations! Your manuscript is now with our production department. 

Kind regards, 

on behalf of

Prof. Ray Borrow 

Academic Editor

PLOS ONE